# Infrared thermochromic antenna composite for self-adaptive thermoregulation

Francisco V. Ramirez-Cuevas[1,2,7], Kargal L. Gurunatha[1,3,7], Lingxi Li[1], Usama Zulfiqar[1], Sanjayan Sathasivam[4,5], Manish K. Tiwari [6], Ivan P. Parkin [5] & Ioannis Papakonstantinou [1] ✉

Self-adaptive thermoregulation, the mechanism living organisms use to balance their temperature, holds great promise for decarbonizing cooling and heating processes. This functionality can be effectively emulated by engineering the thermal emissivity of materials to adapt to background temperature variations. Yet, solutions that marry large emissivity switching ($\Delta\epsilon$) with scalability, cost-effectiveness, and design freedom are still lacking. Here, we fill this gap by introducing infrared dipole antennas made of tunable thermochromic materials. We demonstrate that non-spherical antennas (rods, stars and flakes) made of vanadium-dioxide can exhibit a massive (~200-fold) increase in their absorption cross-section as temperature rises. Embedding these antennas in polymer films, or simply spraying them directly, creates freeform thermoregulation composites, featuring an outstanding $\Delta\epsilon \sim 0.6$ in spectral ranges that can be tuned at will. Our research paves the way for versatile self-adaptive heat management solutions (coatings, fibers, membranes, and films) that could find application in radiative-cooling, heat-sensing, thermal-camouflage, and other.

Self-adaptive thermoregulation is the mechanism living organisms use to maintain their body temperature stable against thermal fluctuations of their surroundings. Researchers have long sought imitating this mechanism, in order to dynamically balance heat influx and outflux in their systems and dampen detrimental temperature swings[1–6]. An efficient way to regulate the temperature of a surface is by controlling the amount of heat it is allowed to radiate. In that respect, both passively and actively controlled radiators have been developed whose emissive power is tuned by some external stimulus (temperature, electrical current, mechanical actuation etc)[3,7–12].

Passive systems are of particular interest as they do not consume energy during operation. Much of the work in this area has focused on utilizing thermochromic materials, i.e., materials undergoing a thermally-driven phase-change around a critical temperature $T_c$[3,6].

Some prominent examples include metal-oxides (e.g., $VO_2$)[7,8,13], chalcogenides [e.g. $Ge_2Sb_2Te_5$ (GST) and $In_3SbTe_2$ (IST)][14,15], perovskites [e.g., $SmNiO3$ and $(La,Sr)MnO_3$ (LSMO)][11,16–18], liquid crystals and other[3,6].

Thermochromism is corollary to self-adaptive heat emission by way of Kirchoff's law of thermal radiation. This is due to the intrinsic absorption of thermochromic materials undergoing a gradual change throughout the phase-change process. Consequently, a varying emissivity profile manifests, bounded between its asymptotic values at low and high temperatures (also known as cold and hot states). The design variable one seeks to maximize is naturally the hot-to-cold emissivity contrast $\Delta\epsilon = \epsilon_h - \epsilon_c$ ($\epsilon_{h/c}$, emissivity in the hot/cold states). For passive thermoregulation purposes, it is essential that $\Delta\epsilon > 0$ (positive differential)[7,8,13,19]. In that case, when the medium deviates from its

[1]Department of Electronic & Electrical Engineering, Photonic Innovations Lab, University College London, London, UK. [2]Center for Energy Transición (CENTRA), Facultad de Ingeniería y Ciencias, Universidad Adolfo Ibáñez, Santiago, Chile. [3]Centre for Nano and Material Sciences (CNMS), JAIN University, Ramanagara, Bangalore, India. [4]School of Engineering, London South Bank University, London, UK. [5]Department of Chemistry, Materials Chemistry Centre, University College London, London, UK. [6]Department of Mechanical Engineering, Nanoengineered Systems Laboratory, University College London, London, UK. [7]These authors contributed equally: Francisco V. Ramirez-Cuevas, Kargal L. Gurunatha. ✉e-mail: i.papakonstantinou@ucl.ac.uk

transition temperature, a radiative feedback develops forcing it back towards $T_c$. The spectral range for maximizing $\Delta\epsilon$ is usually set to either the Long Wavelength Infrared (LWIR) window (8–15 μm), or the Middle Wavelength Infrared (MWIR) window (3–8 μm). The former is utilized in radiative cooling applications, as it aligns with the blackbody peak at temperatures $25-100$ °C and the atmospheric window (8–13 μm)[20,21], while the latter is common for thermal imaging applications[22,23].

The most common approach in the literature to maximize $\Delta\epsilon$ is based on resonant structures. Several variants of this approach have recently appeared including multilayered Fabry-Perot cavities[7,24–26], metasurfaces[8,19,27–31] or thin films[32]. Whilst some of these systems have demonstrated adequate emissivity switching, scaling them up is still a major challenge. A number of them for example, require specialized equipment and cleanroom environments that increase the final cost[7,8,19,29]. Furthermore, their fabrication typically requires hard substrates (silicon, quartz, sapphire or other) to withstand the elevated temperatures during deposition, or post annealing steps and to ensure uniformity across the whole area[7,19,29].

Overall, all these approaches require permanently attached substrates, which impose constraints making them incompatible with anything but planar surfaces. A recent study utilized thermochromic-shelled particles of spherical geometry to overcome some of these issues[13]. Unfortunately, only modest emissivity modulation ($\Delta\epsilon \sim 0.26$ in the atmospheric window) was demonstrated. This is because, as we show later, spherical geometries are not optimum to maximize $\Delta\epsilon$. In addition, they are not the best design either to economize on material usage.

## Results

Departing from the state-of-the-art, we propose an alternative route to create efficient self-adaptive heat radiators by finetuning the absorption cross-section in infrared thermochromic dipole antennas (Fig. 1a, b). In particular, we establish that: (i) Dipole antennas with high surface area-to-volume ratio (SA:V) is key to achieve large absorption cross-section ratios, of which rod geometries (Fig. 1c) are the best performing. (ii) When dispersed in a host material, the collective action of the antennas macroscopically translates into a composite medium with large emissivity switching (Fig. 1e). (iii).

Only a very small amount of antennas is sufficient to attain large $\Delta\epsilon$. The medium hosting them can thus be ultrathin (<100 μm) and flexible (Fig. 1d) with obvious benefits in terms of cost and design freedom. (iv) Antenna resonances are very sensitive to their geometrical dimensions, a feature that can be exploited for wavelength selective radiators. As proof-of-concept material, we use vanadium-dioxide (VO$_2$), an archetype, strongly correlated thermochromic metal-oxide with a temperature driven Insulator-to-Metal Transition (IMT)[33]. Rod-shaped VO$_2$ antennas were made by hydrothermal synthesis (Fig. 1c), a process allowing accurate control of their length and aspect ratio by carefully tuning the reaction stoichiometry, temperature, pH, and post annealing of the reaction products (Methods). The antennas showed uniform morphology, with a width ($W$) × length ($L$) size distribution of $(0.47 \pm 0.01\,\mu m) \times (14.6 \pm 9.2\,\mu m)$.

The powder X-ray diffraction (PXRD) patterns and X-ray photoelectron spectroscopy (XPS) analysis of the final product confirm the VO$_2$ monoclinic (M) phase formation without any impurities, while differential scanning calorimetry (DSC) measurements revealed $T_c = 70.2$ °C (Supplementary Fig. 8). Flexible composite films were subsequently fabricated by hot-pressing a dry mix of high-density polyethylene powder and VO$_2$ antennas over an aluminum film, the latter acting as a back reflector to prevent radiative thermal exchange with the underlying body (Fig. 1d). The composite film exhibited a dramatic emissivity switching at wavelength, $\lambda = 10$ μm (Fig. 1e), from $\epsilon_c = 0.33$ to $\epsilon_h = 0.88$ ($\epsilon_h - \epsilon_c = 0.56$). The hot/cold phase transition of the averaged emissivity in the atmospheric window (Fig. 1f), shows

$\Delta\epsilon = 0.44$ and a narrow hysteresis width, $\Delta T_{hyst} = 7.7$ °C. With a thickness of <100 μm, the film was also fully flexible (Fig. 1d). First-principle radiative transfer calculations showed very good agreement with experimental results, (Fig. 1e and Methods). We fabricated and tested other samples to ensure repeatability, rendering similar results in terms of $\Delta\epsilon$ and $\Delta T_{hyst}$ (Supplementary Fig. 3). To further assess the thermo-responsive mechanism, we imaged our sample with a LWIR camera and compared against two references; one with constant $\epsilon \approx 0\%$ (aluminium) and another with $\epsilon \approx 100\%$ (Carbon black). The transition of our composite from a low to a high emissivity state at $T \sim 70$ °C is clear (Fig. 1g). Thermochromic antennas are a very versatile starting material from which several end-products may be derived. To prove this point, we developed a second proof-of-concept, whereby a solution of VO$_2$ antennas dispersed in acetone was created that could easily be sprayed directly on any surface (Fig. 1b and S5).

## Optimization pathways for thermochromic antenna composites

To maximize $\Delta\epsilon$, the critical design parameter is the antenna's absorption cross-section ratio, $C_{abs,h}/C_{abs,c}$, where $C_{abs,h}(C_{abs,c})$ is the absorption cross section in the hot(cold) phase. This is illustrated in Fig. 2a, using a simplified model based on Beer–Lambert's law (Supplementary Note 2). This model correlates the $C_{abs,h}/C_{abs,c}$ ratio of individual antennas with the total emissivity $\epsilon_{h/c}$ of the composite. As evidenced in Fig. 2a, the smaller the emissivity in the cold state $\epsilon_c$, the larger the absorption cross-section ratio required to maximize $\Delta\epsilon$. To maintain low values of $\epsilon_c$, the parameters $t_{film}/\Lambda_{abs,c}$ and $\alpha_0 t_{film}$ must be minimized, where $t_{film}$ is the thickness of the film, $\alpha_0$ is the absorption coefficient of the host material, $\Lambda_{abs,c} = V_p/f_v C_{abs,c}$ is the absorption mean-free-path of the cold particles, $V_p$ is the particle volume and $f_v$ is volume fraction.

The results from Fig. 2a helps explain why spherical particles can only ever achieve meagre $\Delta\epsilon$, even though they are some of the most popular geometry in the general VO$_2$ literature[7,34,35]. As seen in Fig. 2b, in the best $C_{abs,h}/C_{abs,c}$ that can be achieved in the LWIR region is $\sim 4$ for optimized VO$_2$ spheres of 2 μm in diameter. For such modest absorption cross-section ratios though, $\Delta\epsilon$ is bounded to <0.35. This situation changes starkly when SA:V increases (Fig. 2c). In the figure, SA:V is increased by deforming the sphere down to a disk and then to a rod, leading to a monotonic enhancement of the absorption cross section ratio. It is important to note that for non-spherical particles, it is the orientation and polarization-averaged absorption cross-sections, $\langle C_{abs,c/h}\rangle$, that needs to be considered, which physically represent the average response of the randomly-oriented antennas in the composite[36]. In the hot phase, an increase in SA:V suppresses the non-radiative modes, resulting in an increase of radiative dissipation and hence an enhancement of $\langle C_{abs,h}\rangle$[37,38]. This is further confirmed by mode decomposition analysis (Supplementary Note 4 and Supplementary Fig. 12), which reveals that the absorption cross section is dominated by the contribution of dipole modes as SA:V increases. On the other hand, the absorption in the cold phase is primarily governed by polarization currents rather than free electrons[39]. At wavelengths $\lambda \in 5 - 14$ μm, cold-phase VO$_2$ features only weak polarization currents[33] and $\langle C_{abs,c}\rangle$ is proportional to the volume of the structure. The structures analyzed in Fig. 2c for example, feature $\langle C_{abs,c}\rangle/V_p \approx 0.2$ μm$^{-1}$ (Supplementary Fig. 11). As a result, $\langle C_{abs,c}\rangle$ now reduces as SA:V increases. The net result is a sharp increase to the absorption cross-section ratio, which for a rod may exceed 200, or >50 times larger than the spherical case.

The magnitude and peak of the absorption cross-section ratio can be finetuned by adjusting its width, $W$ (Fig. 2d), and length, $L$ (Fig. 2e), respectively. For example, the peak of the ratio redshifts from $\lambda = 10$ μm to $\lambda = 5$ μm when $L$ reduces from 2 μm to 1 μm. By reducing $W$ from 200 nm to 50 nm the magnitude of the ratio increases by $\sim 4$. Setting a target at $\lambda = 10$ μm, which is in the middle of the LWIR window, we identify that the largest absorption cross-section ratio is reached

for antennas with $L \approx 2\,\mu m$ and $W \approx 50nm$ (Fig. 2f). As revealed by first principle radiative transfer simulations (Fig. 2g), the emissivity contrast of a composite based on a transparent host ($\alpha_{host} = 0\,mm^{-1}$) and these optimum antennas can reach an impressive $\Delta\epsilon = 0.91$ at an antenna volume fraction of just $f_v = 0.11\%$. In a more conservative scenario based on the dimensions of our synthesized antennas, this limit decreases to $\Delta\epsilon = 0.76$ at $f_v = 0.51\%$ v/v. In both cases, however, only a very small concentration of antennas is required to create an efficient self-adaptive heat radiation system. In the simulations, the refractive index of the transparent host is $N_h = 1.5$, which represent a typical value in traditional polymers, such as, polyethylene, polystyrene, polypropylene, and most acrylics[40]. For a real host material, such as PE ($\alpha_0 \approx 1.3\,mm^{-1}$ in the atmospheric window), the maximum $\Delta\epsilon$ is reduced as $\alpha_0 t_{film}$ increases. The parameter $t_{film}/\Lambda_{abs}$, on the other hand, remains constant in the optimum and conservative scenarios, regardless of the values of $t_{film}$ and $\alpha_0$. Interestingly, the predicted $\Delta\epsilon$ shows little sensitivity to polydispersity (Fig. 1h). Considering a normal particle-size distribution with a coefficient of variation of 2, $\Delta\epsilon$ is reduced only by $\sim 5\%$ and $11\%$ for the optimal and conservative scenario, respectively. In our experiments, the coefficient of variation is <0.63, which implies a 3.5% reduction of $\Delta\epsilon$ due to polydispersity. It is important to note that our design rules are universal and applicable on any thermochromic material exhibiting an IMT. For example, maximum $\Delta\epsilon = 0.84$ and 0.92 are predicted for GST and IST rod composites, respectively (Supplementary Fig. 14).

## Effects of the morphology of thermochromic antennas

Attaining large absorption cross-section ratio is by no means bound to antennas of rod geometry only, adding additional degrees of design freedom. An effective way to tune the geometry of an antenna is by changing the reaction pH, temperature, concentration of reductant or surfactant[41–43]. To demonstrate the versatility of this approach, the

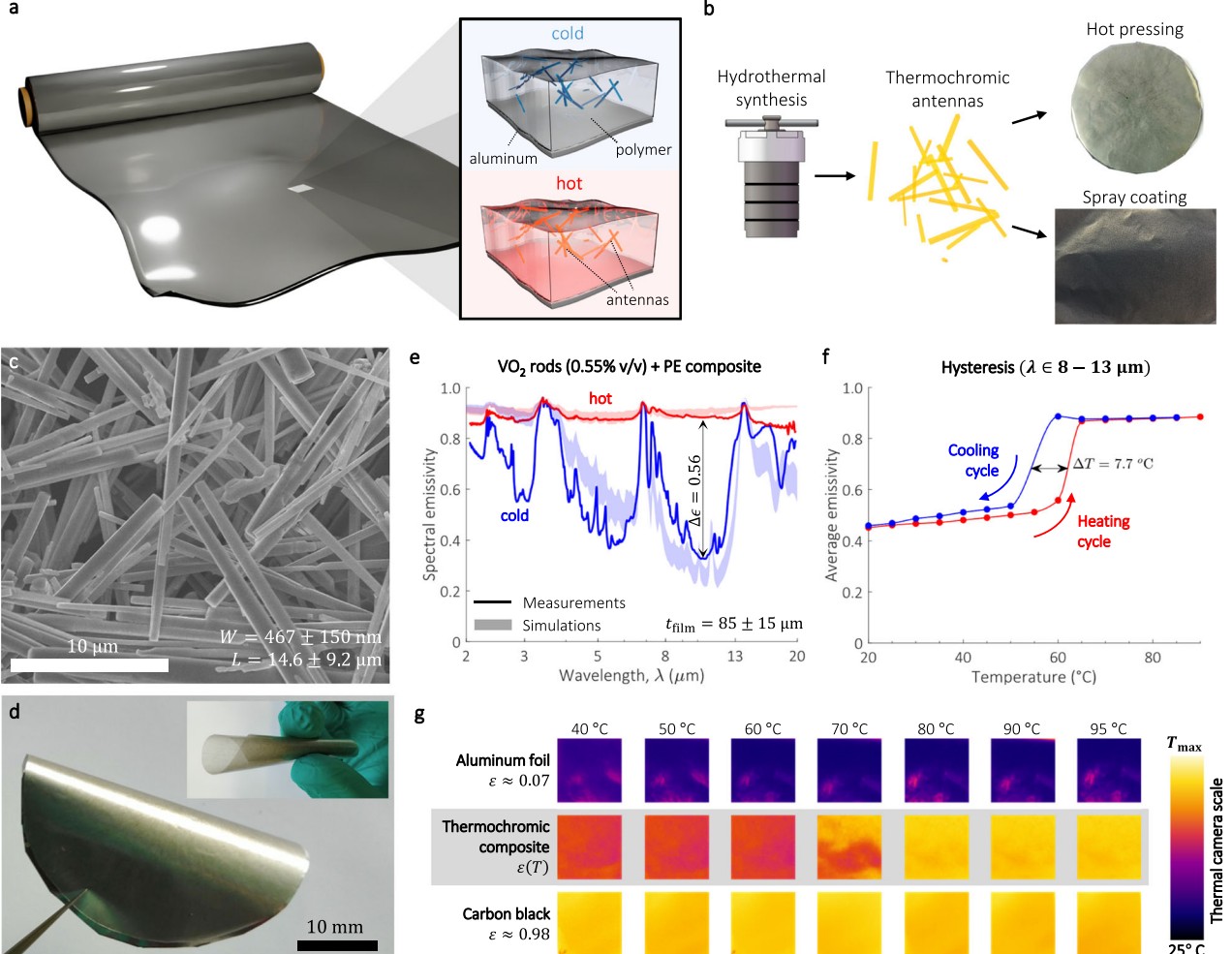

**Fig. 1 | Thermochromic antennas for passive thermoregulation composites.**
**a** Self-adaptive thermoregulation composite film concept consisting of infrared thermochromic antennas embedded in a polymer matrix. **b** Schematic of the fabrication protocol for free-form composites. VO$_2$ antennas were first synthesized by hydrothermal synthesis in a high-pressure autoclave. The polymer films and coatings shown in the photographs were then respectively made by hot pressing and spray coating. A dedicated photograph with a scalebar of a hot-pressed and spray-coated sample is shown in (**d**) and Supplementary Fig. 5, respectively. **c** Scanning electron microscopy (SEM) image of VO$_2$ antennas, showing the width ($W$) and length ($L$) size distribution (Methods). **d** Photograph of composite thermochromic film made by mixing VO$_2$ antennas with polyethylene (0.55% v/v) and hot-pressed against an aluminum foil. The polymer film composite has a thickness of only

$85 \pm 15\,\mu m$ (Supplementary Fig. 4) and is flexible (inset). **e** Measured hot/cold spectral emissivity of the composite film (solid line). The red and blue filled area correspond to numerical simulations using the refractive index of VO$_2$ and PE reported elsewhere (Supplementary Fig. 9)[33,40]. **f** Average emissivity in the atmospheric window, during heating and cooling cycles. **g** Thermal camera images of carbon black ($\epsilon \approx 1$), aluminum ($\epsilon \approx 0$) and thermochromic composite heated from 40 to 95 °C. The scalebar shows the apparent temperature registered by the camera, where $T_{max}$ corresponds to the heating temperature. The switch at ~70 °C of the composite from a low to a high emissivity state is evident. The variation in color tones for the aluminum foil sample are due to wrinkles on the surface, which cause non-uniform temperature distribution under the thermal camera. Source data are provided as a Source Data file.

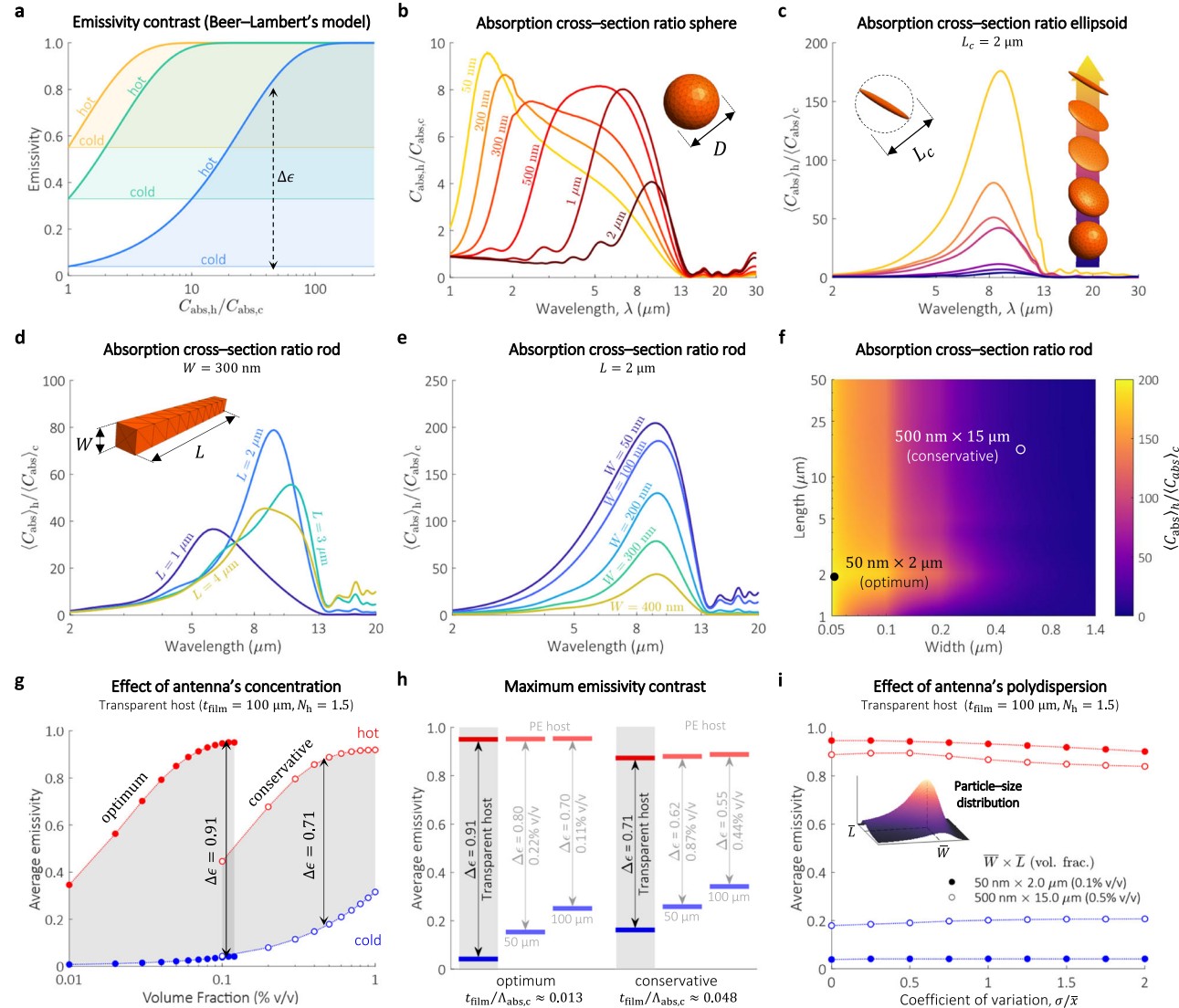

**Fig. 2 | Computational analysis of thermochromic VO₂ antennas and passive thermoregulation composite. a** Emissivity contrast of a composite based on Beer–Lambert's law as a function of the antenna absorption cross-section ratio. **b** Absorption cross-section ratio of $VO_2$ spheres as a function of their diameter, $D$. **c** Absorption cross-section ratio of $VO_2$ ellipsoids for increasing surface-area-to-volume (SA:V) ratios. The lengths of two axes of the ellipsoid are gradually reduced to 100 nm, while the other axis is kept at a length, $L_c = 2\,\mu m$. The absorption cross-section ratio is based on orientation-averaged values, $\langle C_{abs} \rangle$. **d, e** Spectral absorption cross-section ratio of a $VO_2$ rod for different lengths ($L$) and widths ($W$), respectively. The corresponding $\langle C_{abs} \rangle_h$ and $\langle C_{abs} \rangle_c$ are shown in Supplementary Fig. 13. **f** $VO_2$ rod absorption cross-section ratio at $\lambda = 10\,\mu m$. The corresponding $\langle C_{abs} \rangle_h$ and $\langle C_{abs} \rangle_c$ are shown in Supplementary Fig. 13. The filled and open circles indicate, respectively, the rod dimensions for maximum ratio (optimum) and those

observed in our experiments (conservative). **g** Hot/cold average emissivity (atmospheric window) of a composite film based on a transparent host and $VO_2$ rods with optimum and conservative dimensions. The phase transition region is highlighted by the gray area. **h** Maximum $\Delta\epsilon$ (averaged over the atmospheric window) of the composite film shown in (**g**). The maximum $\Delta\epsilon$ of composites based on a PE host of 50 μm and 100 μm thickness are also shown, indicating the volume fractions for maximum contrast. The parameter $\Lambda_{abs,c}$ was computed with $C_{abs,c}$ at $\lambda = 10\,\mu m$. **i** Similar to g, but with $\langle C_{abs} \rangle$ weighted by a gaussian particle-size distribution with a variable coefficient of variation ($\sigma/\bar{x}$), where $\sigma$ is the standard deviation and $\bar{x}$ is the mean. The refractive index of PE and $VO_2$ are reported elsewhere (Supplementary Fig. 9)[33,40]. Source data are provided as a Source Data file.

concentration of the surfactant CTAB was modified, leading to sub-micron $VO_2$ nanostars whose emissivity switching was this time optimized for the MWIR window (Fig. 3a). When embedded into a PE matrix film, the $VO_2$ nanostar composite exhibited a maximum $\Delta\epsilon = 0.52$ at $\lambda = 5.4\,\mu m$ (Fig. 3b), and a narrow hysteresis width, $\Delta T_{hyst} = 7.2\,°C$ (Supplementary Fig. 2). The blue shifting to the maximum $\Delta\epsilon$ is attributed to the smaller dimension of the nanostars and agrees with our modeling results (Fig. 3c).

We further explore the role of geometry, by modeling and comparing five different structures with different SA:V, stars, flakes, rods, a spherical core-shell particle composed of a LWIR transparent core and $VO_2$ shell, and a $VO_2$ spherical particle (Fig. 3d–g). Core-shell particles

have been proposed before in the context of emissivity switching and it is interesting to assess how they compare against our dipole antennas[13]. To make the comparison fair, all structures are optimized to exhibit an absorption cross-section maximum at $\lambda = 10\,\mu m$. Other characteristic dimensions, such as the width (200 nm) or the heigh of the stars $H = (500\,nm)$, were set in agreement with the values reported in literature[13,44]. As shown in Fig. 3e, rods show the largest absorption cross-section ratio (132.4), followed by stars (56.7), flakes (52.1) core-shell spheres (9.7), and spheres (3.9). So far, the impact of scattering in the optical properties of the composite films has not been discussed. However, since the critical dimensions of the antennas are comparable with the wavelength of thermal emission, scattering may play an

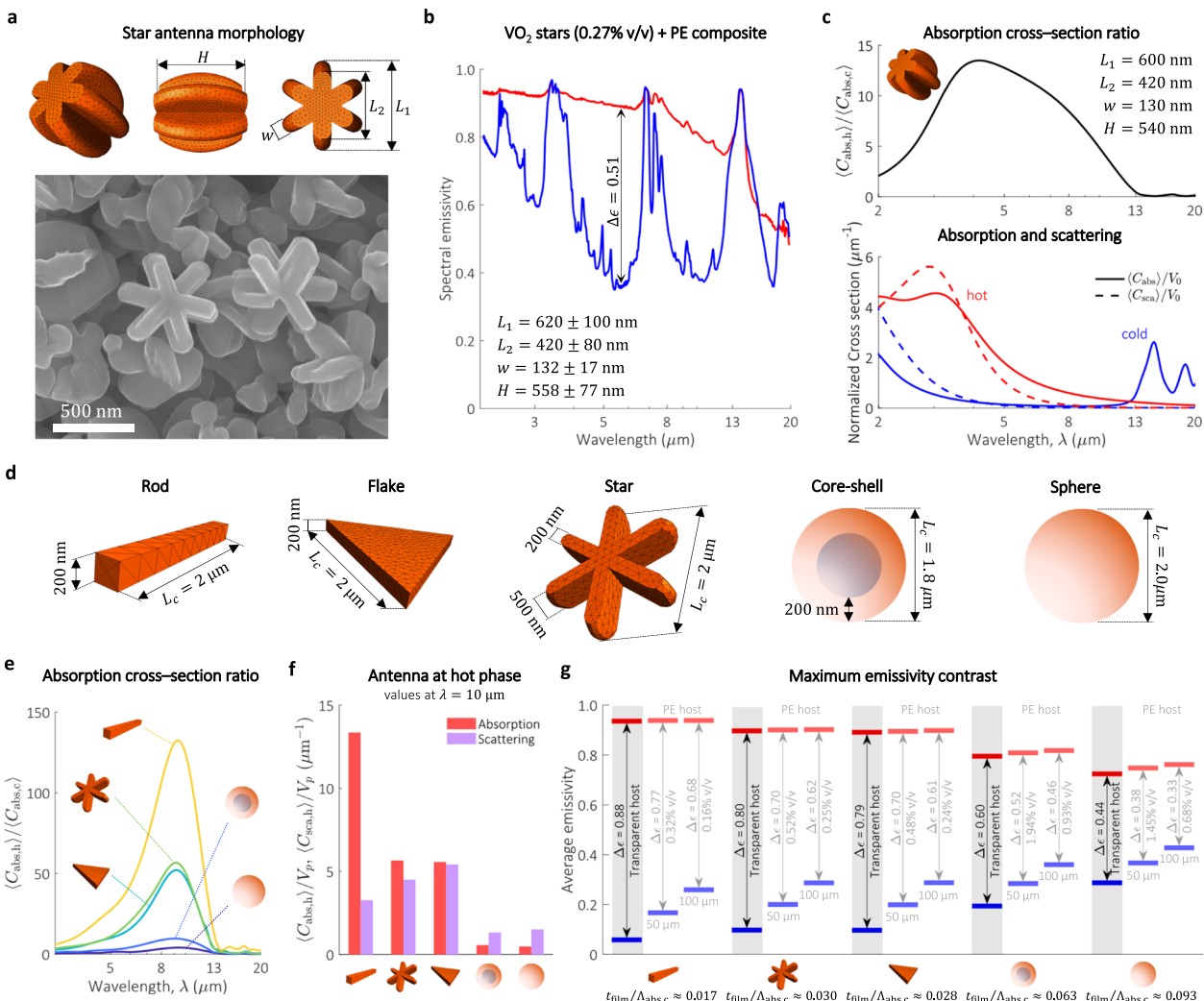

**Fig. 3 | Role of morphology in the performance of thermochromic antennas.**
**a** 3D model used in computational calculations (top) and SEM (bottom) of VO$_2$ nanostars. **b** Measured emissivity of a thermochromic composite based on VO$_2$ stars at hot and cold states. **c** Simulation of absorption cross-section contrast, and normalized scattering and absorption cross-sections of optimum VO$_2$ nanostars (dimensions shown at the top right corner). **d** Three different dipole antenna geometries with large SA:V, a core-shell and a spherical particle, with their optimized characteristic length ($L_c$). **e** Absorption cross-section contrast for the 5 different geometries presented in (**d**). **f** Scattering and absorption cross-sections of the five geometries in the hot state at $\lambda = 10\,\mu m$. The values are normalized to the particle's volume ($V_p$). **g** Maximum $\Delta\epsilon$ (averaged in the atmospheric window) of resulting composite films based on a transparent host. For details, see supplementary note 5 and Supplementary Fig. 15. The maximum $\Delta\epsilon$ of a composite based on a PE host of 50 μm and 100 μm thickness are also shown, indicating the volume fraction for maximum contrast. The parameter $\Lambda_{abs,c}$ was computed with $C_{abs,c}$ at $\lambda = 10\,\mu m$. For core-shell particles, the refractive index of the core is 1.5. The refractive index of PE and VO$_2$ is based on values reported elsewhere (Supplementary Fig. 9)[33,40]. Source data are provided as a Source Data file.

important role. The effect of scattering is accounted for by comparing the (orientation-averaged) absorption and scattering cross sections normalized to the volume of the antenna, $\langle C_{abs}\rangle/V_p$ and $\langle C_{sca}\rangle/V_p$, respectively (Fig. 3f). The normalized cross-sections calculated in Fig. 3f were then used as input to solve the full radiative transfer equation and calculate the emissivity of composite films based on a transparent host (Supplementary Fig. 15). The results are summarized in Fig. 3g, showing the maximum $\Delta\epsilon$ for each morphology. As seen in the figure, scattering might either enhance or reduce $\Delta\epsilon$. If $\langle C_{sca}\rangle/V_p$ is smaller than $\langle C_{abs}\rangle/V_p$, scattering plays a favorable role to enhance $\epsilon_h$ as it increases the optical path in the film[45]. This is evidenced in the results of composites with stars and flakes, whose $\epsilon_h$ is similar to those of rod composites, despite having lower $\langle C_{abs}\rangle/V_p$. Using Beer–Lamberts law (Supplementary Fig. 15), we confirmed that the enhancement of $\epsilon_h$ disappears when the effect of scattering is not present. If $\langle C_{sca}\rangle/V_p$ is larger than $\langle C_{abs}\rangle/V_p$, however, the reflectance of the composite increases, lowering $\epsilon_h$; a feature observed in the results of core-shell and spherical particles. For composites based on a

PE host, we observed similar trends, although with lower $\Delta\epsilon$ values according to the parameter $t_{film}\alpha_0$. Similar to the results in Fig. 2h, the parameter $t_{film}/\Lambda_{abs}$ changes only with the morphology of the particle. An important point to note is that the magnitude of $\langle C_{abs}\rangle_h/V_p$ correlates directly to the antenna concentration $Sf_v$ required to maximize $\Delta\epsilon$. In this regard, rods, flakes and stars composites require at least 5 times less antennas than equivalent core-shell and sphere composites.

In summary, we introduced a new family of infrared thermochromic dipole antennas featuring high absorption cross-section contrast that can be leveraged to construct effective self-adaptive radiators. Antennas can be synthesized in bulk using cheap precursors and then mixed with polymers or other materials lending our technology compatible with numerous scalable industrial manufacturing processes such as film/foil extrusion, fiber extrusion, compression molding, injection molding, dip coating, spray coating, doctor blading, electrospinning and other. In essence, with the right combination of host material and fabrication method, passive thermoregulation products of arbitrary shape and mechanical properties (flexible, rigid,

stretchable, twistable, etc) may be created that can be applied on, virtually, any surface. Our method is also very cost effective. To put this in perspective, a typical hydrothermal reaction in our lab scale autoclaves was sufficient to produce >200 mg of material, when <2.15 mg of rods is required per square meter of composite (Supplementary Fig 1). By exploring the effects of morphology, we demonstrate that antenna resonances are very sensitive to their geometrical dimensions, a feature that can be exploited for wavelength sensitive radiators that switch only in certain spectral bands. For example, monodispersed $VO_2$ stars and rods can be used to cover three different spectral responses, respectively, (i) a radiator switching in the 3–8 μm MWIR window, (ii) a radiator switching in the LWIR 8–15 μm window, and (iii) a broadband radiator responding across both windows. Interestingly, targeting structures with large SA: V can also positively affect the nature of the thermochromic transition. As observed in our experiments, rods and stars tend to grow along well-defined crystallographic planes with high crystallinity[43], resulting in antennas with sharper IMT transition and narrow hysteresis (Fig. 1f), a desirable feature for some applications such as tunable radiative cooling[5,6]. Alternatively, $T_c$ in $VO_2$ can be tuned by W-doping[46], or by combined effect of strain and oxygen vacancies[43] for near room-temperature switching. While rods, stars and flakes designs were investigated in this article, there is no indication these are exclusive. Our design rules are universal extending to other morphologies with large SA:V, as well as other thermochromic materials exhibiting IMT transitions. Furthermore, while only designs with $\Delta\epsilon > 0$ have been discussed, if the antenna concentration exceeds the percolation threshold, this situation may be reversed leading to composites with $\Delta\epsilon < 0$ (negative differential)[47]. In this case, radiative heat losses may stay constant, or even be reversed as the temperature increases[22,48,49]. The apparent temperature of a surface, recorded by an infrared camera, may then be decoupled from its real temperature, potentially creating a thermal camouflage effect[22,23,50].

## Methods

### Sample preparation

Vanadium pentoxide ($V_2O_5$, 99.6%, Sigma Aldrich), hydrazine hydrate solution ($N_2H_4\cdot H_2O$, 85%, Fisher Scientific), Sulfuric acid ($H_2SO_4$, 95%, Sigma Aldrich), Cetyl Trimethy Ammonium Bromide (CTAB, 98%, Sigma Aldrich), Sodium Hydroxide (NaOH, Fisher Scientific), Ethanol ($C_2HOH$, Fisher Scientific), Low Density Polyethylene (LDPE), IRGANOX (BASF) were used as received without further purification.

### Synthesis of vanadium dioxide ($VO_2$) rods (Supplementary Fig. a1).

The synthesis of rods consist of two-step processes as reported earlier[43]. 0.48 g of $V_2O_5$ was dispersed in 10 mL of deionized water with continuous magnetic stirring to form a yellow suspension. 0.75 mL of $H_2SO_4$ was added while heating the suspension at 80 °C. After 15 min, 270 μL of $N_2H_2\cdot H_2O$ was slowly added to the above mixture to form a transparent blue color solution indicating the reduction of $V^{5+}$ to $V^{4+}$. With vigorous stirring, 1.8 mL of CTAB (0.1 M) was added and the solution kept stirring for 1 h. The pH of the resulting solution was then adjusted to 4.0–4.2 by adding NaOH (1 M) dropwise. The brown precipitate was washed with water three times by using a centrifuge and re-dispersed in 19 mL water and transferred to a 45 mL Teflon Autoclave (Parr acid digestion vessel). Hydrothermal reaction was carried out at 220 °C degree for 63 h. The final black precipitate was collected by centrifugation, washed with ethanol, and dried at 80 °C for 1 hr. Thermal transformation of $VO_2$ (A) to $VO_2$ (M) was carried out in a vacuum tube furnace at 550 °C, 0.2 mbar for 1 h.

### Synthesis of Vanadium dioxide ($VO_2$) stars (Supplementary Fig. b1).

The synthesis of stars follows a similar hydrothermal procedure to than for rods. However, the synthesis involved a slightly higher volume of $N_2H_2\cdot H_2O$ (330 μL), a reduced volume of CTAB (300 μL at 0.1 M concentration), and a higher reaction temperature (235 °C degree for

63 h). Under these reaction conditions, $VO_2$ (M) stars were formed in a single step. The resulting stars were collected through centrifugation, washed with ethanol, and dried at 80 °C for 1 h.

All hydrothermally synthesized samples were washed thoroughly with ethanol to remove excess of capping agent and reducing agent.

### Sample Characterization and optical measurements

Morphology and size of the $VO_2$ samples were obtained using a field-emission Scanning Electron Microscope (SEM), JEOL JSM-6701F instrument with an accelerating voltage of 5–10 KeV. For SEM imaging, a small portion was drop-casted on silicon wafer followed by room temperature drying. The average particle size was measured through SEM images using ImageJ software[51]. The crystallographic phase identification of $VO_2$ samples and phase transformation with temperature were determined by PXRD, using a STOE SEIFERT diffractometer with angular range of 2° <2θ < 45° and a Mo K-alpha X-ray radiation source. XPS was carried out on a Thermo Scientific K-alpha photoelectron spectrometer with a dual beam charge compensation system using monochromatic Alkα radiation. High-resolution scans were recorded for the principal peaks of C (1 s), V (2p) and O (1 s) at a pass energy of 50 eV. The binding energies were calibrated with respects to O1s peak at 530.0 eV. All peak fittings were carried out using CasaXPS software. Differential scanning calorimetry (DSC) analysis of as synthesized $VO_2$ samples were performed on a DSC instrument from Mettler Toledo, where experiments were carried out between 25 and 200 °C under nitrogen atmosphere with a heating ramp of 5 °C/min and cooling ramp of 0.5 °C/min. Thermal camera images of $VO_2$ antenna composites on aluminum foil were taken in FLIR A655c High resolution LWIR camera at room temperature and high temperature. The absorption spectra of $VO_2$ antenna composites were measured by Fourier transform Infrared spectroscopy (SCHIMADZU IRTracer −100) with a gold mid-IR integrating sphere (Pike Technologies), and a custom-built thermoelectric temperature controller.

### $VO_2$ polymer composite preparation

A known amount of $VO_2$ (M) rods or stars (0.002g–0.02 g) was mixed with 0.8 g LDPE and 0.01 g of antioxidant (IRGANOX) through solid grinding. This mixture was then poured b/w aluminum plates and subjected to compress molding at 180 °C/350 bar for 3 min. The thickness of the polymer composite film was measured through electronic digital micrometer and cross-sectional SEM which showed average thickness of 80 μm (Supplementary Fig. 4). The measured emission spectra of the composite films with different rod(star) volume fractions are shown in Supplementary Fig. 1a2 and 3(Supplementary Fig. 1b2 and 1b3), and Fig. 1e(3b) of the main text. The volume fraction was estimated from the $VO_2$/LDPE weight ratio and the densities of $VO_2$(M) (4.230 g/cm³)[52] and LDPE (0.925 g/cm³).

### $VO_2$ composite spray coating

The $VO_2$ powder was grounded using mortar and pestle and mixed with acetone to prepare a 2 wt.% suspension. The mixture was left in an ultra-sonication bath for 4 h to achieve a homogenous suspension. Later, the suspension was sprayed on aluminum foil using a spray gun (DeVilbiss DAGR Airbrush, Fluid tip: 0.35 mm, Operating pressure: 1.3–3.5 bar, 10 ml suspension for $2 \times 2$ cm² sample) from 15 cm at an angle of 45°. The temperature of the aluminum substrate was maintained at 50 °C to ensure the rapid evaporation of any residual solvent and the formation of a smooth film. Finally, the composite samples were prepared by dip-coating the spray-coated $VO_2$ film in 2 wt% polyethylene solution in Toluene (Supplementary Fig. 5).

### Orientation-averaged scattering simulations

Orientation-and-polarization-average light scattering simulations were performed using the SCUFF-EM[53] application AVESCATTER[54]. SCUFF-

EM, is an open-source software for electromagnetic simulations, based on the Boundary Elements Method. The meshing of the objects is based on triangular panels and was carried by GMSH[55]. The volume of each structure was numerically computed by GMSH. The number of mesh elements of each structure was determined by performing convergence analysis through mesh refinement, ensuring that the results varied by no more than an absolute tolerance of 0.05. The results for the last step of mesh refinement are summarized in the table below:

**First principle radiative energy transfer simulations**

Radiative transfer simulations were performed by mc-photon, our open-source software for Monte Carlo simulations of unpolarized light[56]. The code has been validated previously[36]. The Monte Carlo algorithm consist on simulating the trajectories of many individual photons as they interact with particles and interfaces, until they are either, absorbed by particles or exit the simulation domain. The initial condition of each photon is given by the position and direction of the light source. At each simulation step, the optical path ($\Lambda_{photon}$) and fate of a photon is estimated by selecting the shortest path between the particle's scattering ($\Lambda_{sca}$) and absorption ($\Lambda_{abs}$), the absorption of the host ($\Lambda_{host}$), or diffraction ($\Lambda_{Fresnel}$), where:

$$\Lambda_{sca} = -\frac{V_p}{f_v \langle C_{sca} \rangle} \ln \xi, \qquad (1)$$

$$\Lambda_{abs} = -\frac{V_p}{f_v \langle C_{abs} \rangle} \ln \xi, \qquad (2)$$

$$\Lambda_{host} = -\frac{\lambda}{4\pi \kappa_{host}} \ln \xi, \qquad (3)$$

and $\Lambda_{Fresnel}$ is given by the shortest distance between the photon and an interface. In the equations above, $\xi$ is a random number between 0 and 1, and $\kappa_{host}$ is the imaginary part of the refractive index of the host.

If diffraction occurs ($\Lambda_{photon} = \Lambda_{Fresnel}$), the photon is either reflected or transmitted by random selection, with the probabilities of each event proportional to the respective energy flux defined by Fresnel laws. If the photon is absorbed by a particle ($\Lambda_{photon} = \Lambda_{abs}$) or the host material ($\Lambda_{photon} = \Lambda_{host}$), the event is terminated, and the simulation continues with a new photon at the initial conditions. For a scattered photon ($\Lambda_{photon} = \Lambda_{sca}$) the new direction, $\theta$, is determined by[57]:

$$\cos \theta = \begin{cases} \frac{1}{2g}\left[1 + g^2 - \left(\frac{1-g^2}{1-g+2g\xi}\right)^2\right] & \text{if } g \neq 0 \\ 2\xi - 1, & \text{if } g = 0 \end{cases} \qquad (4)$$

where $g = \langle \mu_{sca} \rangle$, and $\mu_{sca}$ is the asymmetry parameter.

In all simulations, we considered a slab with large surface area to represent a 2D problem. As a criterion, we selected the smallest surface area by which no photon escapes through the edges. Two large monitors above and below the slab measure the total reflectance and transmittance, respectively. In all the simulation, we considered 1,000,000 photons per wavelength.

For samples featuring polydisperse particle-size distributions, $\langle C_{abs} \rangle$, $\langle C_{sca} \rangle$ and $\langle \mu_{sca} \rangle$ represent the ensemble averaged values weighted by a two-dimensional normal distribution (details in Supplementary Note 3).

## Code availability

The simulation tools used to carry out this study are available open source in Refs. 53,54 and 56.

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

## Acknowledgements

This work was carried out under the framework of the H2020 European Research Council (ERC) Starting Grant IntelGlazing, Grant No. 679891. U.Z., M.K.T., and I.P. would like to thank the Royal Academy of Engi-neering Frontiers Research Grant FF1920182 for funding. L.L. and I.P. would like to thank the UKRI, ERC proof-of-concept grant PolyCool EP/X024482/1 for funding.

## Author contributions

I.P., F.R.C., and K.L.G. conceived the original idea. F.R.C. designed the antennas and performed the simulations. K.L.G. fabricated the antennas and polymer composite films. UZ fabricated spray coated samples. K.L.G., F.R.C., S.S., and L.L. performed all experimental characteriza-tions. I.P., I.P.P., and M.K.T. supervised the research. F.R.C., K.L.G., and I.P. wrote the manuscript with assistance from the other authors. All authors discussed and analyzed the results.

## Competing interests

F.R.C., K.L.G., I.P.P. and I.P. are inventors of a provisional patent application related to this work: "Thermochromic Antenna", appli-cation number application number: EU23386093.1, File Date 09/10/23. The patent covers the use of antennas for passive thermoregulation.
