## [Peer Review File · Nature Communications]

Infrared thermochromic antenna composite for self-adaptive thermoregulationREVIEWER COMMENTS

Reviewer #1 (Remarks to the Author):

The author studied systematically Infrared thermochromic antenna composite for self-adaptive thermoregulation. It is a very good work for VO₂ based composite films for thermoregulation. Unfortunately, the properties of the materials in the manuscript are not outstanding. The authors had better optimize the preparation process to optimize the properties of the material to the same as the existing studies. Besides, this manuscript is very suitable for publication in NC. There are two reminders to the author:

- (1) I do not recommend using a single wavelength emissivity change value to represent the thermal management performance of the material.
- (2) The emittance of Aluminum and carbon black in Fig 1g is not appropriate.

Reviewer #2 (Remarks to the Author):

Review of the manuscript: Infrared thermochromic antenna composite for self-adaptive thermoregulation.

The proposed approach to develop self-adaptive heat radiators by tuning the absorption cross-section of VO₂-based antennas is physically sound and beyond the current state of the art. Authors present a detailed analysis of different geometrical configurations of antennas to maximize their hot-to-cold emissivity contrast. However, the manuscript has some flaws that should be tackled before its possible publication, as detailed in the following comments.

1. On the used material permittivities: The emissivity contrast and the absorption cross-sections are driven by the refractive index of VO₂ and its host material. Authors should therefore justify the reason why they took a real refractive index (1.5) for the host material and explicitly show the one used for VO₂. What are the features of an ideal host material to enhance the hot-to-cold emissivity contrast?

2. On the impact of the VO₂ hysteresis: As is well known, the optical, thermal, and electrical properties of VO₂ take different values during its heating and cooling, at a given temperature. The authors are therefore encouraged to evaluate the impact of this intrinsic property of VO₂ on the hot-to-cold emissivity contrast of the studied antennas.

3. On the Scuff-EM calculations: The light scattering simulations were performed using the software Scuff-EM, which was implemented with a triangular meshing of objects via GMSH. According to Fig. 3d, this meshing tool generated regular triangles, which may not be suitable to describe the simulations in corners. The authors should therefore clarify the used meshing (among the many 2D and 3D possibilities provided by GMSH) and demonstrate that they calculations are independent of it.

Reviewer #3 (Remarks to the Author):

The authors of the manuscript titled “Infrared thermochromic antenna composite for self-adaptive thermoregulation” describe the use of non-spherical VO₂ structures shaped as rods, stars, and flakes to demonstrate IR tunable thermochromic radiators. The featured thermochromic radiators function as IR dipole antennas with high hot-to-cold emissivity contrast and are well suited for passive thermoregulation purposes. The results reported are noteworthy and merit publication. The methodology used is sound and the work presented in the manuscript and supplementary information supports the author’s conclusions. Furthermore, the supplementary information provided is very thorough and includes sufficient detail in order to reproduce the work presented.

In their summary the authors discuss various directions for further work. If allowed by the manuscript length limits, it would be beneficial for those in the field if the authors could further discuss the following:

1. Are there any physical limitations in developing thermoregulation treatments using thermochromic antennas made by hydrothermal synthesis capable of operation across the full IR spectrum (1 to 25 microns)? Could these treatments be produced by building stacks each with a different size of monodispersed VO₂ antennas?
2. For the VO₂ rods or stars exhibiting high crystallinity resulting in sharper IMT transitions would their crystallinity improve with an oxygen post-annealing treatment after completion of the hydrothermal synthesis steps? Would the resulting antennas exhibit higher emissivity contrast given the higher content of the monoclinic VO₂ phase?

RESPONSE TO REVIEWERS' COMMENTS

Dear Sirs/Madams,

We would like to thank you for your thorough work in reviewing our paper and for your constructive comments, which have significantly helped us to improve the quality of our manuscript. We are also grateful to all three of you for praising the novelty of our work and for considering our manuscript as worthy for publication in Nature Communications, contingent upon the implementation of the necessary corrections. Before explicitly addressing your comments, we would like to iterate a few broad points that will set the tone for our more detailed responses:

- 1) **Key-breakthrough:** The key breakthrough of our work is the introduction of composites with randomly dispersed *non-spherical* resonant antennas (e.g. *dipole-rod* or *dipole-star*) for thermal emission control. This represents a fundamental departure from the current state-of-the-art, which (mostly) relies on thin-film Fabry-Perot systems or metasurfaces. Arguably, the most significant benefit of our approach is the simplicity with which end-products, such as polymer foils or coatings, can be fabricated.
- 2) **Efficiency:** Our antenna designs are significantly more efficient compared to some competing designs in the literature based on oversimplified spherical geometries. Such geometries offer minimal control over multipole emissions, thereby limiting their emissivity switching potential. This point is conclusively demonstrated in Fig. 3g, where we show that our rod, star and flake-based antennas can achieve almost complete emissivity switching ($\Delta\epsilon\sim 0.88$). In contrast, spherical or core-shell particles exhibit more limited emissivity switching potential ($\Delta\epsilon < 0.44$ and $\Delta\epsilon < 0.60$ correspondingly).
- 3) **Motivation:** It is critical to emphasize that the primary motivation of this study was to demonstrate a proof-of-concept self-adaptive thermoregulation prototype, rather than to aim for a world record in emissivity switching. While we recognize there is room for improvement, the results presented here are still very competitive (see more details in our response to reviewer 1).
- 4) **Design Universality:** Finally, it is extremely important to convey the message of *design universality* for our thermal antennas. In other words, the proposed technology is not limited to VO₂, which was chosen merely for proof-of-concept purposes. It extends to other materials undergoing an insulator-to-metal-transition (IMT). This argument is supported by our studies on In-Sb-Te (IST) and Ge-Sb-Te (GST), two additional IMT materials with great potential as self-adaptive emitters, as shown in Fig. S12 of our supplementary material.

A point to point answer to your comments will now follow. Please note that comments from reviewers are marked in **blue**, while our answers are in **black**.

Reviewer #1 (Remarks to the Author):

The authors studied systematically infrared thermochromic antenna composite for self-adaptive thermoregulation. It is a very good work for VO₂ based composite films for thermoregulation. Unfortunately, the properties of the materials in the manuscript are not outstanding. The authors had better optimize the preparation process to optimize the properties of the material to the same as the existing studies. Besides, this manuscript is very suitable for publication in NC.

Response: We appreciate the reviewer's positive assessment of our manuscript's suitability for publication in NC. We also acknowledge their suggestion to focus on achieving results comparable to the state-of-the-art in the literature. Our defence is based on three counterarguments:

1. As highlighted in our opening statement above, the most popular thermoregulation technologies involve Fabry-Perot multilayer films and metasurfaces, typically deposited on hard substrates using thin-film deposition processes such as chemical vapour deposition (CVD), atomic layer deposition (ALD), sputtering or other. And while some of these systems have achieved superior emissivity switching indeed, scaling up to the levels required by some applications remains challenging. For instance, radiative cooling, a promising technology to decarbonise cooling in the built environment, would necessitate covering billions of square meters of rooftops with these materials. Our solution, compatible with coatings, paints and polymer membranes (i.e. building materials used extensively in construction), is undoubtedly better suited for such large-scale applications.

2. Our thermochromic antenna composite performs comparably to existing state-of-art solutions with similar characteristics (i.e. compatible with polymer foils). In support of this point, we state that our composite achieved $\Delta\epsilon = 0.44$, while $\Delta\epsilon = 0.26$ was achieved in ref. [13], and $\Delta\epsilon = 0.46$ in ref [7]. The only two pieces of work that exhibited higher emissivity switching were [8] and [31], which were an original design and a follow up by the same group which improved scalability and showed, $\Delta\epsilon = 0.68$ and $\Delta\epsilon = 0.59$ correspondingly. [Note: $\Delta\epsilon$ here is taken as the average value in the atmospheric transparency window, $\lambda \in 8 - 13 \mu\text{m}$ which we calculated ourselves from the data in the references].

3. Referring back to our opening statement, our sole motivation here is to demonstrate a proof-of-concept thermoregulation composite, aiming to spark a new direction of research in self-adaptive thermoregulation. Supported by our comprehensive modelling data (shown in Fig. 3e-g), we anticipate that optimizing antenna dimensions could achieve outstanding emissivity switching, reaching $\Delta\epsilon = 0.8$ and potentially outperforming other systems in the literature (see also response to reviewer 2). However, we would like to reserve such optimization studies, including controlling switching temperature, full-scale systems and outdoors testing, for future research.

There are two reminders to the author:

1. I do not recommend using a single wavelength emissivity change value to represent the thermal management performance of the material.

Response: We agree that a single wavelength emissivity switching value should not represent the material's thermal management performance. Thus, we have recalculated emissivity switching $\Delta\epsilon$ as an average in $\lambda \in 8-13 \mu\text{m}$, corresponding to the LWIR atmospheric transparency window, in all relevant plots. All figures in the main text (Figs. 2g, 2h, 2i and 3g) and Supplementary Materials (Figs. S14 and S15) have been modified accordingly. The figures of the main text are also reproduced below.

From fig. 2

From fig. 3

2. The emittance of aluminium and carbon black in Fig 1g is not appropriate.

Response: Please note that these values were used solely for visually interpreting thermal camera images, and had not propagated into any of the results in this paper. However, at the reviewer's recommendation, we have now corrected the emittance values of aluminium foil (0.07) and carbon black (0.98) according to standard tables. Fig. 1g has also been modified accordingly, and reproduced here:

Reviewer #2 (Remarks to the Author):

The proposed approach to develop self-adaptive heat radiators by tuning the absorption cross-section of VO₂-based antennas is physically sound and beyond the current state of the art. Authors present a detailed analysis of different geometrical configurations of antennas to maximize their hot-to-cold emissivity contrast.

Response: We are very grateful to the reviewer for the positive comments.

However, the manuscript has some flaws that should be tackled before its possible publication, as detailed in the following comments.

1. On the used material permittivities: The emissivity contrast and the absorption cross sections are driven by the refractive index of VO₂ and its host material. Authors should therefore justify the reason why they took a real refractive index (1.5) for the host material and explicitly show the one used for VO₂. What are the features of an ideal host material to enhance the hot-to-cold emissivity contrast?

Response: We will start by addressing the question about the VO₂ refractive index. We agree with the reviewer that it was an omission not to include the VO₂ (n, κ) values in our paper. Even though these values have been widely reported in the literature, it is still beneficial for the readers to have them directly accessible. Therefore, at the request of the reviewer we have now added a figure in the Supplementary Information (Fig. S9) showing the refractive index values of VO₂ used in our simulations. The figure also shows the refractive index used for Polyethylene (PE), which is based on the values for High-Density PE (HDPE). This figure is also reproduced below.

Regarding the question about the refractive index for the ideal host material, it should combine the following three properties to maximize the range of emissivity switching $\Delta\epsilon$:

- a. **The value of n should be as close to 1.0 as possible, to reduce impedance mismatch with air. Otherwise, reflections occur at the air-polymer interface suppressing $\Delta\epsilon$.** Nonetheless, it is not realistic to assume that nanoantennas can simply be suspended in air. HDPE, the host material in our composite foils, has a refractive index of $N \approx 1.5$ in the infrared, similar to polystyrene, polypropylene or acrylics, which are other common polymers in the IR literature. Due to the resemblance in the refractive index of all these materials, we used $N \approx 1.5$ as a baseline refractive index in our optimization calculations, which includes Figs. 2g, 2h, 2i and 3g. Nonetheless, reflecting upon the reviewer comments, we have also added optimization values for composites based on a PE host of thickness 50 μm and 100 μm , which are shown in Figs. 2h and 3g. These figures are also reproduced here.

- b. **The value of κ should be as close to 0 as possible, as increasing this value reduces the emissivity contrast.** Specifically, the absorption coefficient of the host $\alpha_0 = 4\pi\kappa/\lambda$ should be as low as possible.
- c. **For a given value of α_0 , the thickness of the composite film (t) should be as low as possible to enable large emissivity contrast.** In our models, we fixed the thickness to $t = 100 \mu\text{m}$, which was very close to the value obtained with our polymer hot pressing method.

In the revised version of the manuscript, we have updated Fig. 1a reducing the number of variables to specifically show the effects of increasing the absorption of the host. We have also added a few paragraphs to:

- Add a discussion on the properties for an ideal host material.
- Better illustrate the choice of the values $n = 1.5$ and $\kappa = 0$ in the optimization figures.
- Discuss the effect of reducing the thickness of the composite film.

In the section “Optimization pathways for thermochromic antenna composites” (first paragraph)

-To maximize $\Delta\epsilon$, the critical design parameter is the antenna’s absorption cross-section ratio, $C_{\text{abs,h}}/C_{\text{abs,c}}$, where $C_{\text{abs,h}}$ ($C_{\text{abs,c}}$) is the absorption cross section in the hot(cold) phase. This is illustrated in figure 2a, using a simplified model based on Beer-Lambert’s law (Supplementary Note S2.2). This model correlates the $C_{\text{abs,h}}/C_{\text{abs,c}}$ ratio of individual antennas with the total emissivity $\epsilon_{\text{h/c}}$ of the composite. As evidenced in Fig. 2a, the ~~larger~~ smaller the emissivity in the cold state ϵ_c , the larger the absorption cross-section ratio required to maximize $\Delta\epsilon$. To maintain low values of ϵ_c , the parameters $t_{\text{film}}/\Lambda_{\text{abs}}$ and $\alpha_0 t_{\text{film}}$ must be minimized, where t_{film} is the thickness of the film, α_0 is the absorption coefficient of the host material, $\Lambda_{\text{abs,c}} = V_p/f_v C_{\text{abs,c}}$ is the absorption mean-free-path of the cold particles, V_p is the particle volume and f_v is volume fraction.

Also in section “Optimization pathways for thermochromic antenna composites” (third paragraph)

this limit decreases to $\Delta\epsilon = 0.76$ at $f_v = 0.51\%$ v/v. In both cases, however, only a very small concentration of antennas is required to create an efficient self-adaptive heat radiation system. In the simulations, the refractive index of the transparent host is $N_h = 1.5$, which represent a typical value in traditional polymers, such as, polyethylene, polystyrene, polypropylene, and most acrylics.⁴¹ For a real host material, such as PE ($\alpha_0 \approx 1.3 \text{ mm}^{-1}$ in the atmospheric window), the maximum $\Delta\epsilon$ is reduced as $\alpha_0 t_{\text{film}}$ increases. The parameter $t_{\text{film}}/\Lambda_{\text{abs}}$ on the other hand, remains constant in the optimum and conservative scenarios, regardless of the values of t_{film} and α_0 . Interestingly, the predicted $\Delta\epsilon$ shows little sensitivity to

Fig2a was simplified to directly illustrate the effects of the absorption cross-section ratio on $\Delta\epsilon$ for a given value of ϵ_c . The paragraphs above now describe how to control the magnitude of ϵ_c

Fig. 2g and 2i explicitly shows the thickness and refractive index of the host. Fig. 2h compares the maximum $\Delta\epsilon$ of a composite based on a transparent host with that of a PE host with thickness $50\ \mu\text{m}$ and $100\ \mu\text{m}$. The values of $t_{\text{film}}/\Lambda_{\text{abs}}$ for each are also shown for reference.

Similarly, in fig. 3g we repeated the analysis of 2h, now for rod, star, flake, core-shell and spherical particles.

2. On the impact of the VO2 hysteresis: As is well known, the optical, thermal, and electrical properties of VO2 take different values during its heating and cooling, at a given temperature. The authors are therefore

encouraged to evaluate the impact of this intrinsic property of VO₂ on the hot-to-cold emissivity contrast of the studied antennas.

Response: We agree with the reviewer that understanding the behaviour of our system during its phase transition is of profound importance. This is why we measured the emissivity at different temperatures during both heating and cooling cycles for one of our composites containing rods and reported the measurements in Fig 1f. From this graph, important parameters such as the transition temperature, hysteresis width and transition sharpness were experimentally derived. We repeated the measurements for other samples and we found similar behaviour. However, for completeness, we have now included a new hysteresis measurement for a second composite, this time made of stars, in the supplemental material (Fig. S2). The figure is also being reproduced here for reference:

It is worth elaborating further on the hysteresis issue for the reviewer’s benefit. As we discuss in the conclusions section of our paper, our nanoantenna exhibit positive transition characteristics. Specifically we mention, “Interestingly, targeting structures with large SA:V can also positively affect the nature of the thermochromic transition. As observed in our experiments, rods and stars tend to grow along well-defined crystallographic planes with high crystallinity, resulting in antennas with sharper IMT transition and narrow hysteresis (Fig. 1f), a desirable feature for some applications such as tunable radiative cooling.”

3. On the Scuff-EM calculations: The light scattering simulations were performed using the software Scuff-EM, which was implemented with a triangular meshing of objects via GMSH. According to Fig. 3d, this meshing tool generated regular triangles, which may not be suitable to describe the simulations in corners. The authors should therefore clarify the used meshing (among the many 2D and 3D possibilities provided by GMSH) and demonstrate that their calculations are independent of it.

Response: At the core of Scuff-EM it is the Boundary Element Method (BEM). The BEM method computes the internal and scattered electromagnetic fields of particles through fictitious surface currents, whose values are found by satisfying the boundary conditions on the particle’s surface. This implies that only the surface of the object is discretized using 2D elements, significantly reducing the computational cost of the simulation. The software is built to work on triangular elements, whose size must be considerably smaller than the wavelength to ensure accurate results.

In our particular case, all simulated objects are of subwavelength dimensions, which eases constraints on dimensions of each mesh element. However, to address the reviewer’s concern we performed additional meshing tests on every simulated object by comparing the results in our paper with those from a much-refined mesh.

In the revised version of the manuscript, we included a comparison table to show the results of mesh refinement on the three objects simulated in Fig. 3, i.e., rod, star and flake. W

Object	Number of triangular panels		Absolute error					
			$\langle C_{\text{abs}} \rangle / V_p$		$\langle C_{\text{sca}} \rangle / V_p$		$\langle \mu_{\text{sca}} \rangle / V_p$	
	Case 1	Case 2	cold	hot	cold	hot	cold	hot
Rod	88	352	3.174e-02	1.745e-02	4.083e-02	1.644e-02	8.322e-04	8.547e-04
Star	1116	1890	1.303e-03	4.129e-03	1.567e-03	3.956e-03	7.731e-05	2.492e-04
Flake	1160	2576	1.303e-03	4.129e-03	1.567e-03	3.956e-03	7.731e-05	2.492e-04

Lastly, it is worth mentioning that Scuff-EM was not built to retrieve orientation-average absorption and scattering power. Thus, we designed our own tool to accomplish such results, which is based on the functions from Scuff-EM library. This tool, which we named AVESCATTER, is now an additional module of Scuff-EM, and its available open-source (https://github.com/PanxoPanza/scattering_random_orientation.git). As detailed in the supplementary document, there is second step in the simulation that uses monte-carlo radiative transfer using the orientation-average parameters obtained from AVESCATTER as inputs. The monte-carlo simulation software, is also available open-source. The whole methodology to simulate composites with randomly oriented particles of arbitrary shape, was also developed by us and already published in a peer-reviewed journal [Ramirez-Cuevas, F. et al, ACS Photonics 9, 672–681 (2022)]. Thus, we are completely confident of the accuracy of the results shown in this work.

Reviewer #3 (Remarks to the Author):

The authors of the manuscript titled “Infrared thermochromic antenna composite for self-adaptive thermoregulation” describe the use of non-spherical VO₂ structures shaped as rods, stars, and flakes to demonstrate IR tunable thermochromic radiators. The featured thermochromic radiators function as IR dipole antennas with high hot-to-cold emissivity contrast and are well suited for passive thermoregulation purposes. The results reported are noteworthy and merit publication. The methodology used is sound and the work presented in the manuscript and supplementary information supports the author’s conclusions. Furthermore, the supplementary information provided is very thorough and includes sufficient detail in order to reproduce the work presented.

Response: We are very grateful to the reviewer for their positive comments.

In their summary the authors discuss various directions for further work. If allowed by the manuscript length limits, it would be beneficial for those in the field if the authors could further discuss the following:

1. Are there any physical limitations in developing thermoregulation treatments using thermochromic antennas made by hydrothermal synthesis capable of operation across the full IR spectrum (1 to 25 microns)? Could these treatments be produced by building stacks each with a different size of monodispersed VO₂ antennas?

Response: We thank the reviewer for their insightful recommendation. As discussed in the paper, the antenna response is sensitive to its geometrical dimensions (e.g. the length and diameter of a rod), which can be controlled during the hydrothermal synthesis method. In principle, this method can effectively tune the emissivity peak to cover a wide part (or all) of the IR spectrum. Evidence of this is presented in Fig. 1d, where the peak heat emission for single VO₂ rods shifts from ~6 μm for a 1 μm long rod to ~13 μm for a 3 μm rod. However, tuning the antenna dimension alone will unfortunately not achieve thermoregulation across the full 1-25 μm spectrum, even with multiple stacks, as the switching range for VO₂ antennas is fundamentally limited by the refractive index and extinction coefficient (n, k) of the material. In the metallic (hot) phase, VO₂ has relatively high n and k values, resulting in strong absorption. Conversely, in the insulating (cold) phase, the n and k values decrease, allowing more heat to transmit through, thereby building up large emissivity switching. Unfortunately, this effect diminishes beyond the 13 μm range, where the n and k values of both hot and cold phases converge, ceasing emissivity switching beyond this range.

To extend this operation across the full IR spectrum, different thermochromic materials can be used (which is possible as noted in our opening statement on design universality). These materials however should exhibit insulator-to-metal transition and significantly different n and k values across the whole 1-25 μm range. The two materials Ge₂Sb₂Te₅ (GST) and In₃SbTe₂ (IST), which we theoretically examined in the supporting information (Fig S12), have the potential to demonstrate such wider emissivity switching, though this remains to be proven in practice.

2. For the VO₂ rods or stars exhibiting high crystallinity resulting in sharper IMT transitions would their crystallinity improve with an oxygen post-annealing treatment after completion of the hydrothermal synthesis steps? Would the resulting antennas exhibit higher emissivity contrast given the higher content of the monoclinic VO₂ phase?

Response: Annealing the hydrothermally synthesized VO₂ sample indeed leads to enhanced crystallinity. This phenomenon was extensively investigated in our previous study published in *Advanced Functional Materials* in 2020 (30, 2005311). In that study, we found that the thermal annealing process, conducted under varying vacuum conditions, induced the phase transformation of metastable VO₂(A) into VO₂(M2), (M2+M3), (M1), and higher valence V₆O₁₃ phases. Additionally, distinct multiple phase transitions, characterized by both increased and suppressed transition temperatures (T_c), were observed depending on the annealing temperature and the level of oxygen vacancies present. The latent heat of phase transition, measured by differential scanning calorimetry (DSC), also showed significant improvement with the enhancement of sample crystallinity.

Our findings indicate that thermal annealing of the hydrothermally synthesized sample should be carefully controlled. When exceeded a certain threshold in our experiments, thermal deformation of the particles was observed, which decreased crystallinity and caused abrupt changes in the phase transition temperature.

Additionally, thermal annealing of the sample will result in uniform and well-defined structure with fewer surface defects and primarily affects the material's electronic structure resulting in distinct band structures. Increase in crystallinity also reduces phonon scattering and increases thermal conductivity and, indirectly, thermal emissivity.

However, it is important to note that the relationship between crystallinity and emissivity can be complex and may vary depending on the specific application and the characteristics of the nanoparticles. Therefore, detailed experimental studies are necessary to determine the exact impact of improved crystallinity on emissivity contrast in a particular system. We would like to reserve such a detailed study for a future publication.

On behalf of the submitting team

Ioannis Papakonstantinou

REVIEWERS' COMMENTS

Reviewer #1 (Remarks to the Author):

The author has answered my question well, but there are still two suggestions before publication. One is that the resolution of the pictures in the paper is too low, and it is suggested that the author provide high-quality pictures; the other is that the Al samples in Figure 1g are not uniform under the infrared thermal imager, and it is suggested that the author explain or replace it.

Reviewer #2 (Remarks to the Author):

The authors have properly addressed my previous comments, so I recommend the publication of their work.

Reviewer #3 (Remarks to the Author):

The authors have addressed all the reviewer concerns and answered their questions quite well. The revised manuscript is suitable for publication.

RESPONSE TO REVIEWERS' COMMENTS

Dear Sirs/Madams,

We would like to thank you for your thorough work in reviewing our paper and for your constructive comments, which have significantly helped us to improve the quality of our manuscript. We are also grateful to all three of you for praising the novelty of our work and for considering our manuscript as worthy for publication in Nature Communications.

Reviewer #1 (Remarks to the Author):

The author has answered my question well, but there are still two suggestions before publication. One is that the resolution of the pictures in the paper is too low, and it is suggested that the author provide high-quality pictures; the other is that the Al samples in Figure 1g are not uniform under the infrared thermal imager, and it is suggested that the author explain or replace it.

Response: The resolution of the three figures has been fixed according to the guidelines stated by Nature Communications. The non-uniform colouring of the Al samples was due to some small wrinkles in the sample that caused non-uniform temperature distribution. A brief paragraph clarifying this issue was included in the caption of Figure 1. We included other changes to the figure caption to address the comments of the editors. A revised version of the entire caption is reproduced below:

Fig. 1. Thermochromic antennas for passive thermoregulation composites. **a.** Self-adaptive thermoregulation composite film concept consisting of infrared thermochromic antennas embedded in a polymer matrix. **b.** Schematic of the fabrication protocol for free-form composites. VO₂ antennas were first synthesized by hydrothermal synthesis in a high-pressure autoclave. The Polymer-polymer films and coatings shown in the photographs were then respectively made by hot pressing and spray coating respectively. A dedicated photograph with a scalebar of a hot-pressed and spray-coated sample is shown in (d) and Supplementary Fig. 5, respectively. **c.** Scanning electron microscopy (SEM) image of VO₂ antennas, showing the width (W) and length (L) size distribution (Methods). **d.** Photograph of composite thermochromic film made by mixing VO₂ antennas with polyethylene (0.55% v/v) and hot-pressed against an aluminum foil. The polymer film composite has a thickness of only $85 \pm 15 \mu\text{m}$ (Supplementary Fig. 4) and is flexible (inset). **e.** Measured hot/cold spectral emissivity of the composite film (solid line). The red and blue filled area correspond to numerical simulations using the refractive index of VO₂ and PE reported elsewhere (Supplementary Fig. 9).^{33,40} **f.** Average emissivity in the atmospheric window, during heating and cooling cycles. **g.** Thermal camera images of carbon black ($\epsilon \approx 1$), aluminum ($\epsilon \approx 0$) and thermochromic composite heated from 40°C to 95°C. The scalebar shows the apparent temperature registered by the camera, where T_{max} corresponds to the heating temperature. The switch at $\sim 70^\circ\text{C}$ of the composite from a low to a high emissivity state is evident. The variation in color tones for the aluminum foil sample are due to wrinkles on the surface, which cause non-uniform temperature distribution under the thermal camera.

Reviewer #2 (Remarks to the Author):

The authors have properly addressed my previous comments, so I recommend the publication of their work.

Response: We are very grateful to the reviewer comments, and for recommending our work for publication.

Reviewer #3 (Remarks to the Author):

The authors have addressed all the reviewer concerns and answered their questions quite well. The revised manuscript is suitable for publication.

Response: We are very grateful to the reviewer comments, and for recommending our work for publication.

On behalf of the submitting team

Ioannis Papakonstantinou
Professor of Photonics and Nanofabrication
Head, Photonic Innovations Lab

Author Team: Dr Francisco V. Ramirez-Cuevas, Dr Kargal L. Gurunatha, Dr Lingxi Li, Dr Usama Zulfiqar, Dr Sanjayan Sathasivam, Prof Manish K. Tiwari, Prof Ivan P. Parkin and Prof Ioannis Papakonstantinou.